# Are Time Series Foundation Models Ready for Oil and Gas Drilling Anomaly Detection?

Soumyadipta Sengupta [1]   Abdallah Benzine [1]   Sebastiaan Buiting [1]   Ankush Mishra [1]

## Abstract

Anomaly detection of multivariate time-series data is an important component of oil and gas drilling anomaly detection, including early identification of hookload and torque events associated with stuck-pipe risk. Although recent Time Series Foundation Models (TSFM) have shown strong forecasting performance, their value for industrial anomaly detection tasks remains unclear. In this paper, we evaluate two forecasting-based TSFM models, Chronos-2 and Toto, against lightweight CNN, CNN-FNO, and MLP baselines for hookload and torque anomaly detection.

Our experiments show that Chronos-2 obtains the highest F1 score on both tasks. However, the gain is task-dependent. Domain specific forecasting-based finetuning gives only a small improvement on hookload, while it improves Chronos-2 substantially on torque anomaly detection. We also find that a 5M-parameter CNN-FNO reaches near-parity with the 120M-parameter Chronos-2 on the torque task, with much lower inference time, but does not close the gap on hookload.

These results suggest that newer TSFM models can be useful for drilling anomaly detection, particularly when combined with domain-specific forecasting based finetuning. At the same time, their advantage is not uniform across anomaly types, and simpler architectures remain strong candidates for real-time deployment. The study highlights the need to evaluate TSFM not only by accuracy, but also by latency, and cost.

[1] AIQ Intelligence, Abu Dhabi, United Arab Emirates. Correspondence to: Soumyadipta Sengupta <soumyadipta.sengupta@aiq.ae>.

*Proceedings of the 43rd International Conference on Machine Learning*, Seoul, South Korea. PMLR 306, 2026. Copyright 2026 by the author(s).

## 1 Introduction

Deep learning approaches have shown strong performance for time-series anomaly detection, but often require extensive architecture-specific experimentation and large labeled datasets for effective deployment in industrial settings (Yan et al., 2024; Pang et al., 2021). These challenges have motivated recent interest in time-series foundation models (TSFM) to reduce task-specific engineering and labeled data requirements.

Time Series Foundation Models have demonstrated strong zero-shot and fine-tuned capabilities on standard forecasting benchmarks. However, adapting these models to specialized industrial domains often exposes a representational gap between general pretraining objectives and dense downstream tasks. In oil and gas drilling—where sensor data is noisy and irregular—prior generations of TSFM like Chronos (Ansari et al., 2024), Moirai(Woo et al., 2024), MOMENT (Goswami et al., 2024) as well as classical methods like MINIROCKET (Dempster et al., 2021), XGBOOST (Chen & Guestrin, 2016) have consistently struggled to outperform lightweight, fully convolutional networks (CNNs) in industrial classification (Buiting et al., 2024) and segmentation tasks (Khaouja et al., 2025). These studies leave open whether the newest forecasting-centric models change this picture, whether an explicit domain-adaptation stage can close the gap, and whether the same conclusions hold for *torque* anomalies, which neither examined.

In this work, we systematically evaluate whether newer, forecasting-centric TSFM, namely Chronos-2 (Ansari et al., 2025) and Toto (Cohen et al., 2024), can bridge this gap for multivariate anomaly detection. We also introduce an intermediate self-supervised step: fine-tuning the TSFM on a massive corpus (50 billion data points) of unannotated rig data via a forecasting objective prior to anomaly detection adaptation. Our central result is that this domain-adapted Chronos-2 substantially outperforms the CNN on torque (0.69 vs. 0.42 F1), driven mainly by the adaptation stage ($0.47 \rightarrow 0.69$), whereas on hookload the gain is modest and largely due to pretrained weights rather than adaptation ($0.51 \rightarrow 0.52$).

We benchmark these foundation models against lightweight

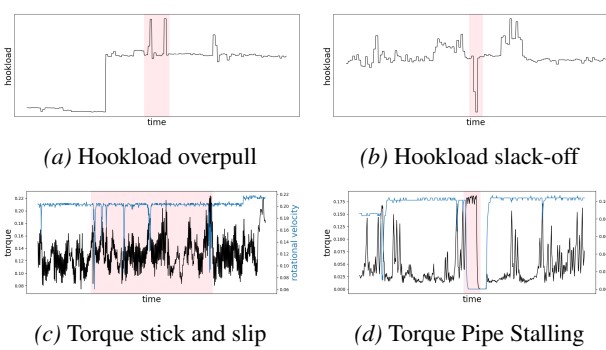

*(a) Hookload overpull*  *(b) Hookload slack-off*

*(c) Torque stick and slip*  *(d) Torque Pipe Stalling*

*Figure 1.* Hookload and Torque Anomalies

MLP, CNN (Buiting et al., 2024; Khaouja et al., 2025) and CNN-FNO (Fourier Neural Operator) (Wen et al., 2022) (Li et al., 2020) architectures. Through ablation studies on weight initialization and dataset scaling, we isolate the contributions of pretrained weights and architectural choices (encoder-centric vs. decoder-centric). Ultimately, this paper provides a cost-benefit analysis tailored to industrial applications, questioning whether the performance gains of 120-million-parameter foundation models justify their exponential computational overhead compared to highly optimized, lightweight alternatives.

We focus on anomaly detection in hookload and torque signals, which are key indicators of major drilling risks such as stuck pipe events (See Figure 1). Each stuck pipe event can result in lost revenues in the order of tens of thousands of dollars. (Muqeem et al., 2012). Hookload and torque anomalies are very rare events occurring on average once every five days per rig lasting for a few minutes only. Hookload anomalies such as overpull and slack-off appear as sharp variations, while torque anomalies like stick-slip and pipe stalling exhibit oscillatory patterns. However, these patterns cannot be identified from a single signal in isolation; annotation requires joint interpretation of multiple sensors and operational context.

A key challenge is that hookload anomalies are scale-dependent: similar magnitude variations may be normal at larger depths but anomalous at shallower depths. Instance normalization commonly used in TSFM disregards this scale change effect which can limit their ability to distinguish true anomalies from normal variations.

## 2  Methodology

We compare an MLP, a CNN, a CNN-FNO (Fourier Neural Operator) model, and two forecasting-based TSFM models, Chronos-2 and Toto, for hookload and torque anomaly detection posed as a dense segmentation task. Each model takes a multivariate sensor window as input and predicts a class label at each timestep.

The CNN baseline (Figure 5) follows the lightweight con-

volutional segmentation model used in prior drilling work (Buiting et al., 2024; Khaouja et al., 2025) . The CNN-FNO model combines this CNN branch with a multilayer Fourier Neural Operator (FNO) branch in parallel (Figure 6). The CNN branch is intended to capture local, high-frequency patterns, while the FNO branch captures lower-frequency structure. The parameters used for the CNN and CNN-FNO models are in Table 2 and 3 respectively. The MLP had 2 hidden layers of dimensions 64 and 128 and an output dimension of 64 to keep it similar to the CNN with 64 features.

Chronos-2 and Toto are originally forecasting models. We first evaluate them by attaching a simple segmentation head to their encoders and fine-tuning them fully on the labeled anomaly datasets. We also tested a 2 stage training for Chronos-2, where we finetuned on forecasting on an unlabeled large dataset and then used this finetuned model encoder to finetune on anomaly labeled smaller dataset.

We also perform ablation studies on Chronos-2. First, we compare random initialization against published pretrained weights to measure the impact of pretraining. Second, we compare Chronos-2 before and after forecasting-based domain adaptation. Third, we vary the amount of labeled training data used for segmentation on both hookload and torque tasks.

### 2.1  Segmentation and Loss

The input $\mathbf{X} \in \mathbb{R}^{B \times features \times T}$ is processed by each model (MLP, CNN, CNN-FNO) or TSFM encoder to extract latent features $\mathbf{H} \in \mathbb{R}^{B \times N_{\text{out\_feats}} \times T}$. A convolutional segmentation head maps $\mathbf{H}$ to class probabilities $\hat{\mathbf{Y}} \in \mathbb{R}^{B \times C \times T}$. The labeled anomaly targets are of shape $\mathbf{Y} \in \{1, \ldots, C\}^{B \times T}$, we optimize the model using multiclass focal loss, where $B$ is the batch size, $T$ is the length of the time series, and $C$ is the number of classes.

## 3  Dataset

All raw time series were sampled at 1 s. A large unlabeled forecasting corpus of around 3000 time series (11 features, around 1.25 million hours in total) was used for domain adaptation. Two smaller labeled datasets were used for segmentation-based anomaly detection: a hookload set with 167448 train, 7779 validation, and 46955 test windows of length 32768 s (4 s subsampling), and a torque set with 94576 train, 1580 validation, and 13186 test windows of length 8192 s. The hookload test set is the same as that used by Khaouja et al. (2025) (re-windowed); see Appendix D. Feature lists, strides, and forecasting setup are given in Appendix B.

## 4 Training

### 4.1 Domain Adapted Finetuning

Before anomaly-detection finetuning, Chronos-2 was domain-adapted by self-supervised forecasting on the large unlabeled rig corpus, using its native weighted quantile loss. The full procedure and hyperparameters are given in Appendix A.

### 4.2 Supervised Training for Segmentation

All models were then trained for segmentation with a multi-class focal loss that down-weights easy examples (focusing parameter $\gamma = 3$, class weights $\alpha = [1, 10]$ for the negative and positive classes). The full loss expression and optimization settings (learning rates, batch sizes, epochs) are given in Appendix A; notably, the foundation models used batch size 2 and the lightweight models 16.

## 5 Evaluation

We evaluate performance using the IoU-based F1 score. A prediction counts as a True Positive (TP) if its IoU with a ground truth segment exceeds a given threshold of 0.1. We chose a small IoU threshold since we were primarily interested in detection. Unmatched predictions and ground truths constitute False Positives (FP) and False Negatives (FN), respectively. All experiments ran on an NVIDIA V100 (32 GB); times are reported per window. Chronos-2 and Toto used batch size 2 (memory-limited) while the lightweight models used 16; since per-window time falls with batch size, the reported latency gap is an upper bound on the lightweight models' advantage.

## 6 Results and Discussion

### 6.1 Main Findings

Figure 2 compares all models on hookload and torque anomaly detection. Chronos-2 outperforms all other models on both tasks, after forecasting-based domain adaptation, reaching 0.52 on hookload and 0.69 on torque. The gain is task-dependent. On hookload, Chronos-2 improves over CNN and CNN-FNO by about 13 percentage points, likely benefiting from its longer effective context. On torque, CNN-FNO reaches 0.68 F1, almost matching Chronos-2. This shows that TSFM models can help, but their benefit depends on the type of anomaly.

Lightweight baselines remain strong. CNN outperforms Toto on both tasks, and CNN-FNO is especially competitive on torque despite being much smaller than Chronos-2.

**Relation to prior work.** On the same hookload test set, Khaouja et al. (2025) report a best F1 of 12.6% (Moment-Large) and 12.2% for a from-scratch CNN, but at a stricter

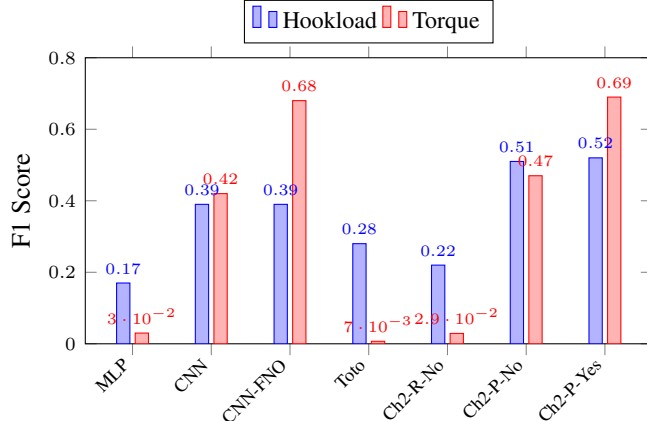

*Figure 2.* F1 score for hookload and torque anomaly detection across model configurations, ordered by increasing architectural complexity. Ch2 = Chronos-2; R = random initialization; P = published weights; Yes/No indicates forecasting-based domain adaptation. *Best configuration for CNN-FNO with 4 s and 1 s subsampling rate for hookload and torque shown*

IoU threshold of 0.5 with macro-F1. Our higher absolute scores reflect our more lenient detection threshold (0.1) and micro-F1, not superior capability. But, most importantly we notice for the first time a TSFM (Chronos-2) far exceeds the CNN on hookload, whereas they found CNNs competitive compared to a variety of TSFMs (Moment, Chronos-1, TimesFM etc.); we attribute this mainly to the newer pretrained weights of Chronos-2. (Appendix D).

### 6.2 Effect of Pretraining and Domain Adaptation

Chronos-2 benefits strongly from published pretrained weights. On hookload, F1 increases from 0.22 with random initialization to 0.51 with pretrained weights. On torque, F1 increases from 0.029 to 0.47.

Forecasting-based domain adaptation has a small effect on hookload, improving F1 from 0.51 to 0.52. On torque, the effect is much larger, improving F1 from 0.47 to 0.69. This aligns with the anomaly structure: torque anomalies are oscillatory and well matched by a forecasting objective, whereas the sharp, scale-dependent hookload transients—whose absolute scale is removed by instance normalization—benefit little.

Toto received neither the random-initialization nor the domain-adaptation ablations applied to Chronos-2, because its published-weight F1 was already below the CNN baseline (0.28 hookload, 0.007 torque); we flag this asymmetry as a limitation. We likewise did not domain-adapt the lightweight baselines, whose much smaller size makes large gains from self-supervised adaptation unlikely.

With only one decoder-based model (Toto) and one encoder-based model (Chronos-2), we cannot establish a general architectural rule; we note only that Toto's weakness is

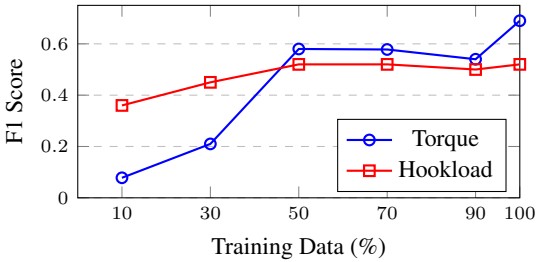

*Figure 3.* F1 score vs. training data percentage for torque and hookload anomaly detection using Chronos-2 after forecasting-based domain adaptation.

directionally consistent with Khaouja et al. (2025), who found encoder-based models outperformed decoder-only ones across six TSFMs on segmentation.

### 6.3 CNN-FNO analysis

The multi layer FNO model was added as a parallel model to the CNN model (Figure 6). The FNO typically captures low frequency behavior in data. The torque anomalies had observable low frequency oscillations (Figure 1) while the hookload anomalies were sharp peaks or troughs. This might explain why the CNN-FNO model performed well for torque but not hookload anomalies.

### 6.4 Effect of Labeled Data Size

Figure 3 shows the effect of reducing the labeled training set for Chronos-2 after forecasting-based adaptation. Hookload saturates around 50% of the labeled data, with little improvement afterward. The advantage is that performance reached maximum with less amount of labeled data which justifies use of TSFM(s).

Torque behaves differently. Performance remains low with small subsets, improves around 50–70%, and increases again with the full dataset. This suggests that torque anomaly detection may still benefit from more annotated examples.

### 6.5 Accuracy and Inference-Time Tradeoff

Figure 4 compares F1 score and inference time. For inference and training, Nvidia V100 (32GB) GPUs were used. Chronos-2 gives the best accuracy, but its inference time is about one order of magnitude higher than CNN and CNN-FNO. For hookload, this cost may be justified because Chronos-2 gives a clear F1 improvement. For torque, CNN-FNO is more attractive because it nearly matches Chronos-2 while being much faster and smaller.

**Limitations.** We evaluate two foundation models in depth rather than broadly; confirming our findings on newer architectures (e.g. TimesFM, MOIRAI 2.0, TabPFN-TS, TiRex)

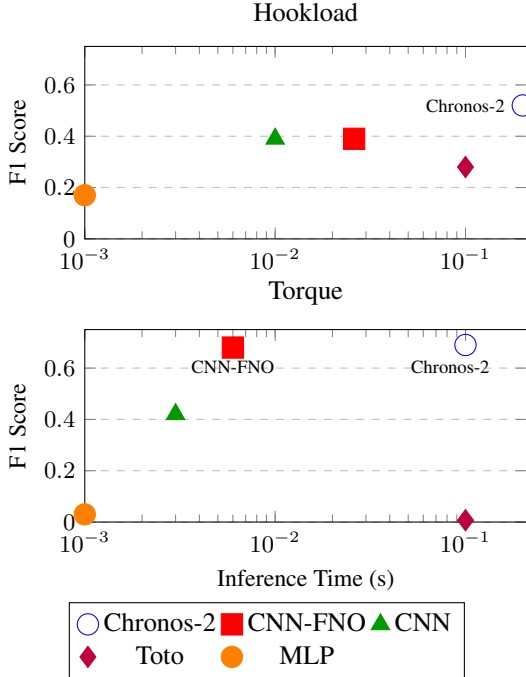

*Figure 4.* Inference time vs. F1 score on hookload and torque anomaly detection. *Best configuration for CNN-FNO with 4 s and 1 s subsampling rate for hookload and torque shown*

and against stronger anomaly-detection baselines, reporting confidence intervals for these rare events, and a dedicated ablation isolating instance normalization on scale-dependent hookload anomalies are left to future work.

## 7 Conclusion

Newer class of TSFM like Chronos-2 show promising results when applying them on the difficult problem of multivariate anomaly detection posed as segmentation. Performance of such models was boosted by large self supervised forecasting objective finetuning before finetuning on the actual classification objective. A combination of CNN-FNO architecture achieved near parity with the Chronos-2 model on the torque anomaly detection task with an order of magnitude fewer parameters and inference time. For real time inference both high accuracy and low inference time is of importance. Therefore, further research into more architectures and TSFM quantization will be performed.

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

# A    Training Details

**Domain-adapted finetuning.**    Chronos-2 was finetuned on a forecasting objective on the large unlabeled dataset to test whether downstream anomaly-detection performance could be further improved. A chunk of 4 files/time series was selected randomly from the 2900 time series; from such a chunk, random windows of length 32768 s and 8192 s with 4 s subsampling and no subsampling were used for hookload and torque respectively. Each chunk received 50 training steps (a forward and backward pass at batch size 2). One epoch covered all 2900 files in chunks of 4 time series ($\approx$72500 windows per epoch). Finetuning ran up to 20 epochs with the native weighted quantile loss, learning rate $1 \times 10^{-6}$ (Chronos-2 default), and patience 6 epochs (tolerance 0.001 on validation loss).

**Supervised segmentation training.**    We optimized the segmentation model(s) using the multiclass focal loss:

$$\mathcal{L} = -\frac{1}{BT} \sum_{b=1}^{B} \sum_{t=1}^{T} \sum_{c=1}^{C} \alpha_c \, y_{b,t,c} \left(1 - \hat{p}_{b,t,c}\right)^{\gamma}$$
$$\cdot \log(\hat{p}_{b,t,c}),$$
$$\gamma = 3, \quad \alpha = [1, \, 10] \text{ for negative and positive classes}$$

$y_{b,t,c} \in \{0, 1\}$ is the one-hot ground-truth label, $\hat{p}_{b,t,c}$ is the predicted softmax probability for class $c$, and $\alpha_c$ is a class-weighting factor; the loss down-weights easy examples and emphasizes harder ones. Learning rates were $1 \times 10^{-6}$ (Chronos-2 default), $5 \times 10^{-4}$ (Toto default), and $1 \times 10^{-4}$ (MLP, CNN, CNN-FNO). Batch size was 2 for Chronos-2 and Toto and 16 for the other models. Training ran up to 100 epochs with patience 6 epochs (tolerance 0.001 on validation F1).

# B    Dataset Details

**Forecasting (domain-adaptation) dataset.**    Around 3000 time series spanning multiple days each at 1 s sampling ($\approx$1.25 million hours total), with 11 features: hook load, rotation velocity, stand pressure, weight on bit, torque, flow rate, flow out percentage, bit depth, block position, hole depth, and tank volume. 2900 series were used for training and 100 for validation (validation stride 1000 s). The forecast horizon was 512 s with 4 s subsampling for hookload and 128 s with no subsampling for torque.

**Hookload anomaly dataset.**    167448 train, 7779 validation, and 46955 test windows of length 32768 s at 4 s subsampling, stride 600 s, using 8 features: hook load, rotation velocity, stand pressure, torque, flow rate, bit depth, block position, hole depth. For CNN-FNO testing only, an additional 1 s-subsampled version was used.

**Torque anomaly dataset.**    94576 train, 1580 validation, and 13186 test windows of length 8192 s with no subsampling, stride 600 s, using 6 features: hook load, rotation velocity, torque, bit depth, block position, hole depth.

# C    Table of all results

*Table 1.* Performance metrics, execution times, and parameter counts for various models on Hookload and Torque anomaly detection. All Hookload models use a subsampling interval of 4 s. The best CNN-FNO configuration is displayed. Times are per window on an NVIDIA V100 (32 GB); Chronos-2 and Toto run at batch size 2 (memory-limited) and the MLP, CNN, and CNN-FNO at batch size 16. As per-window time decreases with batch size, the reported times favor the lightweight models, so the latency gap is an upper bound on their per-window advantage.

| Model | Initial weights | Finetuned on forecasting | Prediction | F1 (micro) | Recall (micro) | Train time/ window (s) | Inference time/ window (s) | Params. (M) |
|---|---|---|---|---|---|---|---|---|
| Linear Classification | Random | No | Hookload | 0.17 | 0.71 | 0.001 | 0.001 | 0.017 |
| CNN | Random | No | Hookload | 0.39 | 0.59 | 0.02 | 0.01 | 1 |
| CNN-FNO | Random | No | Hookload | 0.39 | 0.59 | 0.052 | 0.026 | 2 |
| Toto | Published weights | No | Hookload | 0.28 | 0.56 | 0.11 | 0.1 | 151 |
| Chronos-2 | Random | No | Hookload | 0.22 | 0.64 | 0.75 | 0.2 | 120 |
| Chronos-2 | Published weights | No | Hookload | 0.51 | 0.64 | 0.75 | 0.2 | 120 |
| **Chronos-2** | **Published weights** | **Yes** | **Hookload** | **0.52** | **0.62** | **0.75** | **0.2** | **120** |
| Linear Classification | Random | No | Torque | 0.03 | 0.45 | 0.001 | 0.001 | 0.017 |
| CNN | Random | No | Torque | 0.42 | 0.97 | 0.003 | 0.003 | 0.5 |
| CNN-FNO | Random | No | Torque | 0.68 | 0.97 | 0.006 | 0.006 | 5 |
| Toto | Published weights | No | Torque | 0.007 | 0.29 | 0.1 | 0.1 | 151 |
| Chronos-2 | Random | No | Torque | 0.029 | 0.066 | 0.26 | 0.1 | 120 |
| Chronos-2 | Published weights | No | Torque | 0.47 | 0.80 | 0.26 | 0.1 | 120 |
| **Chronos-2** | **Published weights** | **Yes** | **Torque** | **0.69** | **0.94** | **0.26** | **0.1** | **120** |

# D    Relation to Khaouja et al. (2025)

Khaouja et al. (2025) previously benchmarked foundation models on hookload segmentation using the same test data, reporting a best F1 of $12.6\%$ (univariate) and $10.6\%$ (multivariate) for Moment-Large, with a from-scratch CNN reaching $12.2\%$ (univariate). Although the test set is shared, a direct numerical comparison with our hookload results ($0.52$ for domain-adapted Chronos-2, $0.39$ for the CNN) is confounded by the scoring protocol: (i) we count a detection as correct at an IoU threshold of $0.1$, whereas they require IoU $> 0.5$, a $5\times$ stricter criterion; (ii) we report micro-F1 while they report macro-F1; and (iii) windowing differs ($32{,}768$ s windows at 4 s subsampling vs. 512-step windows at stride 128). Our higher absolute scores therefore reflect the more lenient threshold, not superior detection. Qualitatively, our finding that a foundation model can *exceed* the CNN on hookload (by $\sim13$ points) differs from their conclusion that CNNs match or beat foundation models; we attribute this primarily to the newer model and its pretrained weights rather than to domain adaptation, which improved hookload only marginally ($0.51 \to 0.52$). We present these as complementary observations on shared data under different evaluation criteria. A fully protocol-matched comparison—re-scoring our predictions at IoU $0.5$ with macro-F1—is a straightforward extension for future revisions.

# E    CNN and FNO architectures

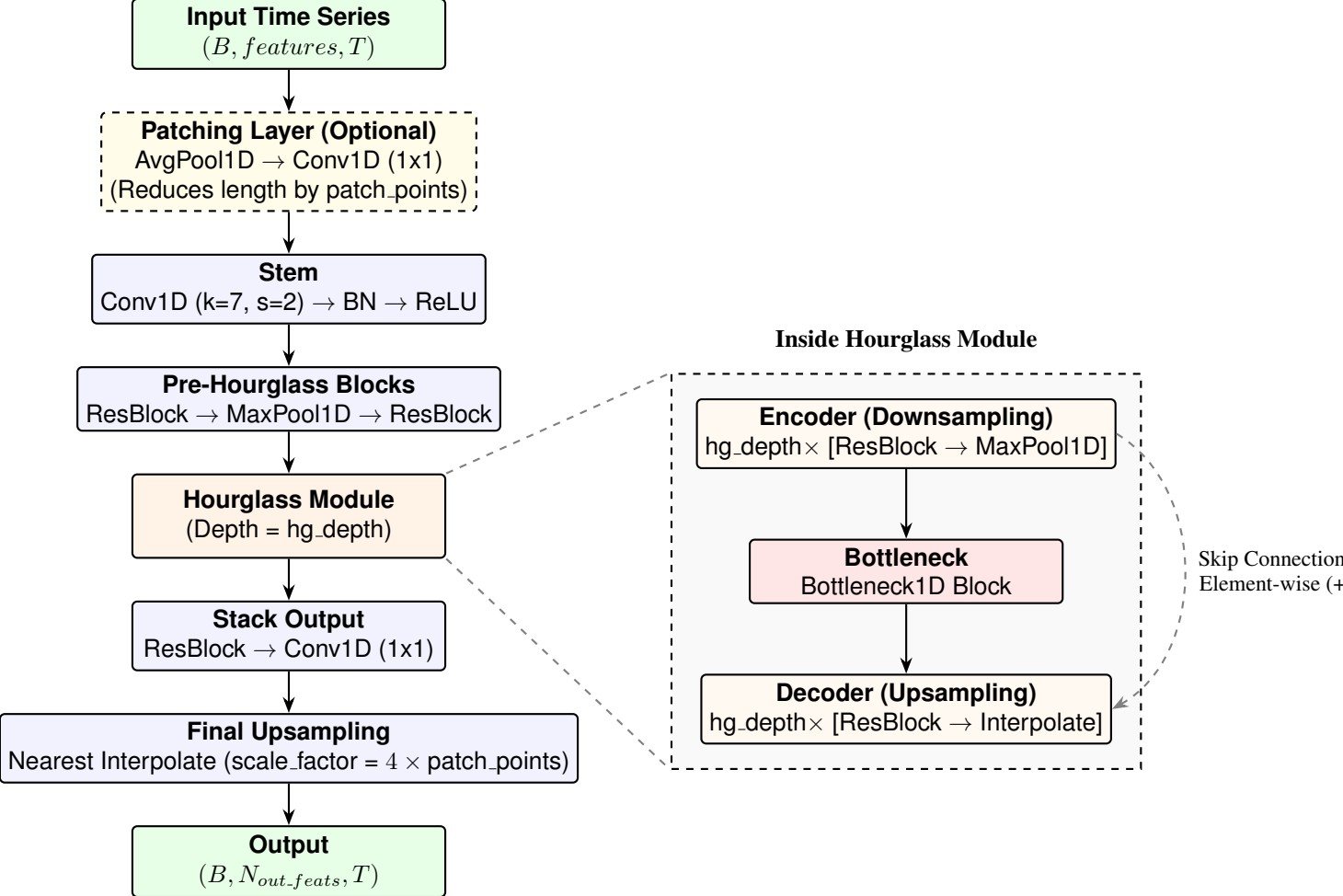

*Figure 5.* CNN (Hourglass1D) architecture for long-sequence time-series segmentation.

*Table 2.* Hourglass1D hyperparameters for Hookload and Torque models

| Parameter | Hookload | Torque |
|---|---|---|
| features | 8 | 6 |
| $N_{out\_feats}$ | 64 | 64 |
| hg_depth | 8 | 2 |
| num_stacks | 1 | 1 |
| num_blocks | 1 | 1 |
| use_attention | False | False |
| max_attention_window | 1024 | 1024 |
| recursive_hourglass | True | True |
| return_intermediate_representation | False | False |
| patch_points | 1 | 1 |
| patch_out_feats | None | None |

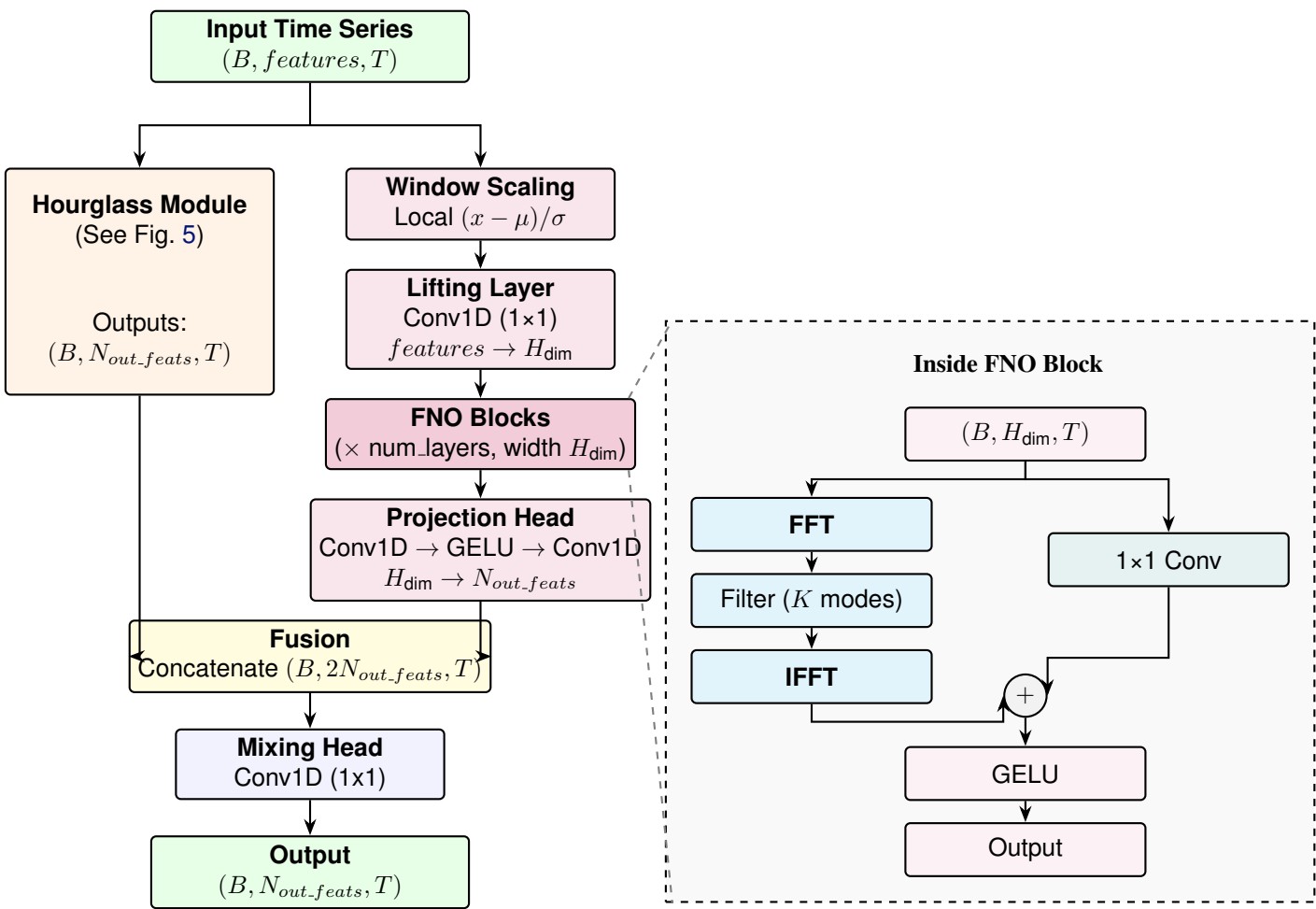

*Figure 6.* Hybrid CNN-FNO architecture combining local (Hourglass) and global (Fourier) representations.

## F    CNN-FNO ablation

| PREDICTION | WINDOW | SUBSAMPLE | HIDDEN DIM (FNO) | MODES | FNO LAYERS | F1 | RECALL |
|---|---|---|---|---|---|---|---|
| HOOKLOAD | 32768 | 4 | 64 | 32 | 8 | 0.39 | 0.59 |
| HOOKLOAD | 32768 | 4 | 64 | 16 | 4 | 0.40 | 0.50 |
| HOOKLOAD | 32768 | 4 | 128 | 32 | 8 | 0.31 | 0.38 |
| HOOKLOAD | 32768 | 4 | 128 | 16 | 4 | 0.34 | 0.57 |
| HOOKLOAD | 32768 | 1 | 64 | 16 | 4 | 0.37 | 0.48 |
| HOOKLOAD | 32768 | 1 | 64 | 32 | 4 | 0.40 | 0.52 |
| **HOOKLOAD** | **32768** | **1** | **128** | **32** | **8** | **0.41** | **0.53** |
| TORQUE | 8192 | 1 | 64 | 32 | 8 | 0.32 | 0.98 |
| TORQUE | 8192 | 1 | 64 | 16 | 4 | 0.64 | 0.81 |
| **TORQUE** | **8192** | **1** | **128** | **32** | **8** | **0.68** | **0.97** |

*Table 3.* CNN-FNO configurations and performance across Hookload and Torque prediction tasks. For reference only CNN model with subsampling of 1 s on hookload task had f1 score of 0.42 and recall of 0.59

Hookload model training windows for models apart from CNN-FNO used a subsampling of 4 s. For CNN-FNO, no subsampling was also modeled along with the 4 s subsampling rate. Subsampling was reduced to check if missing frequencies would improve the FNO performance. In addition, hidden dimension, number of the lowest modes and number of FNO layers were varied for both tasks. For torque anomaly detection task, the FNO model was achieved near parity with Chronos-2 for a particular set of FNO model parameters. However for hookload task, the performance of these models stayed below the Chronos-2 encoder based model irrespective of subsampling rate.

### F.1    CNN-FNO Analysis

CNN-FNO performs similar to Chronos-2 on torque, but not on hookload. A likely explanation is that torque anomalies, such as stick-slip and pipe stalling, contain oscillatory patterns. The FNO branch can capture lower-frequency context, while the CNN branch captures local changes.

Figure 7 shows that FNO weights increase across low-frequency modes and then saturate for both hookload and torque models. This suggests that the FNO branch learns low-frequency structure. However, this does not improve hookload performance, likely because hookload anomalies are sharper transient events where low-frequency context alone is not sufficient.

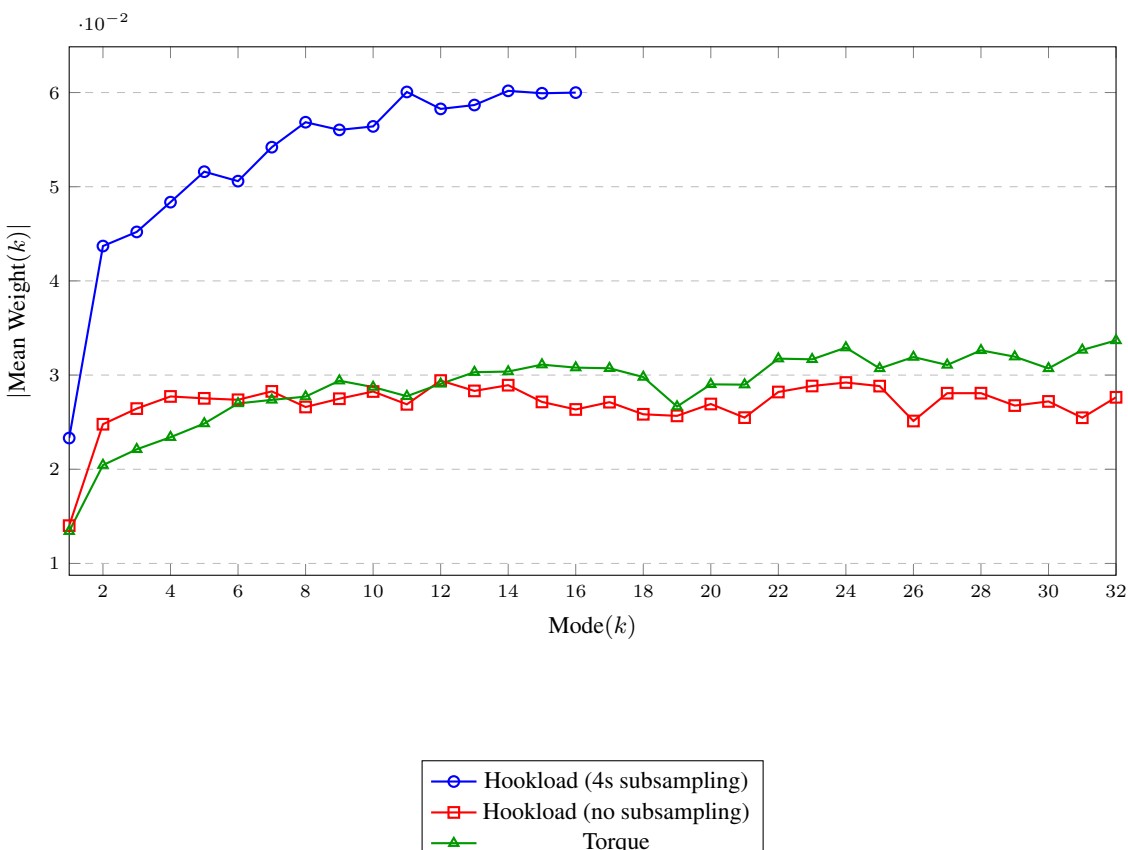

*Figure 7.* Absolute value of FNO layer weights across modes for hookload with and without subsampling, and torque.

