# OpenReview forum: "Are Time Series Foundation Models Ready for Oil and Gas Drilling Anomaly Detection?"
_ICML.cc/2026/Workshop/FMSD — FMSD @ ICML 2026 Poster_

### Official Review · Reviewer_PBAo · 2026-05-13

**Rating:** 7
**Confidence:** 4

**Review:**

## Summary
The authors evaluate forecasting-based Time Series Foundation Models (TSFMs), specifically Chronos-2 and Toto, against lightweight baselines (MLP, CNN, and a hybrid CNN-FNO) for anomaly detection in oil and gas drilling. Focusing on hookload and torque anomalies, the authors frame the problem as a dense segmentation task. The results show that while domain-adapted Chronos-2 achieves the highest F1 scores across both tasks, lightweight models like CNN-FNO can reach near-parity on specific anomalies (such as torque) with an order of magnitude lower inference time and significantly fewer parameters. The paper concludes that while TSFMs are useful when combined with domain-specific finetuning, simpler architectures remain highly competitive for real-time industrial deployment.

## Strengths
- Relevance to Workshop: The paper is highly aligned with the goals of the Foundation Models for Structured Data (FMSD) workshop, specifically addressing the "Real-World Applications" topic by applying structured data foundation models to anomaly detection task.
- Thorough Empirical Evaluation: The authors provide a robust cost-benefit analysis tailored to industrial applications. Comparing a 120-million-parameter foundation model against highly optimized lightweight alternatives (like the 5M-parameter CNN-FNO) adds significant practical value.
- Practicality and Trade-offs: Evaluating models not just by their accuracy (F1 score), but also by their latency (inference time) provides crucial context for real-world, real-time anomaly detection. Furthermore, the ablation studies regarding weight initialization, data scaling, and forecasting-based domain adaptation offer concrete insights into how TSFMs behave in specialized domains.

## Areas for Improvement
- Limited Model Variety: The primary penalty for this submission is the relatively small number of TSFMs evaluated. The foundation model evaluation focuses almost exclusively on Chronos-2 and Toto. The analysis could be significantly strengthened by extending the evaluation to other architectures to see if the findings generalize.
- Missing Hardware Specifications: While the paper heavily emphasizes inference time and computational tradeoffs, it lacks details regarding the hardware environment used for the experiments.

## Detailed Comments
- As suggested in the areas for improvement, incorporating additional architectures like TabPFN-TS, Flowstate and Tirex would make the comparative analysis more comprehensive.
- The paper hypothesizes that the poor performance of Toto might indicate that encoder-based TSFM models are better suited for dense segmentation tasks than decoder-based forecasting models. Testing more models (like TimesFM and MOIRAI 2.0) would help validate whether this is a general architectural rule or an artifact of Toto specifically.
- Given the strong focus on accuracy vs. inference-time tradeoffs, explicitly stating the hardware setup (e.g., CPU/GPU type, RAM) used for the experiments is essential. Because inference latency is highly hardware-dependent, including this information will greatly improve the reproducibility of the study and better contextualize the real-time deployment claims.

## Justification of Score
This paper is well done and perfectly aligned with the experimental track of the FMSD workshop. It rigorously tackles real-world challenges utilizing structured data foundation models and highlights critical tradeoffs between model scale, accuracy, and latency. The slight penalties are the limited selection of foundation models tested and the omission of exact hardware specifications used for the benchmarks.

---

### Official Review · Reviewer_dV59 · 2026-05-21
**Useful industry benchmark update, but novelty is incremental given prior work in same area**

**Rating:** 5
**Confidence:** 4

**Review:**

This paper evaluates whether newer TSFMs — Chronos-2 and Toto — can outperform lightweight CNN and CNN-FNO baselines for anomaly detection in oil and gas drilling data, focusing on hookload and torque signals. A two-stage approach is introduced where Chronos-2 is first finetuned on 50 billion unlabeled drilling data points before supervised adaptation. Chronos-2 achieves the best F1 on both tasks, but a 5M-parameter CNN-FNO nearly matches the 120M-parameter Chronos-2 on torque while being significantly faster.

Strengths
1. Practical cost-benefit framing. Explicitly comparing accuracy against inference time and parameter count is the right lens for industrial deployment. Figure 4 is one of the clearest contributions.
2. Domain adaptation at scale. Finetuning on 50 billion unlabeled drilling data points before supervised adaptation is new. The finding that this substantially boosts torque but barely helps hookload is practically informative.
3. Encoder vs decoder insight. Confirming that decoder-based TSFMs (Toto) underperform encoder-based ones (Chronos-2) for dense segmentation is a useful finding, now extended to anomaly detection specifically.
4. CNN-FNO mechanistic explanation. The spectral analysis in Figure 7 provides a reasonable explanation for why CNN-FNO works on torque but not hookload, grounding the result in signal structure.

Areas for Improvement
1. Incremental novelty. This is the third paper from what appears to be the same group. Buiting et al. (2024) already showed CNNs outperform TSFMs on drilling classification, and Khaouja et al. (2025) already studied hookload segmentation specifically with similar conclusions. The novelty rests primarily on newer models and the torque task, which needs to be stated more transparently.
2. No comparison to Khaouja et al. on hookload. Since that paper already benchmarked foundation models on hookload, the current findings should be directly compared. This comparison is entirely absent.
3. Asymmetric treatment of Toto. Chronos-2 benefits from 50 billion data points of domain adaptation while Toto receives none. This is not a fair comparison and needs justification.
4. Single metric, no significance testing. Reporting only IoU-based F1 without confidence intervals is insufficient for rare event detection with high class imbalance.

Detailed Comments

The introduction should explicitly state what Buiting et al. and Khaouja et al. left unanswered, and frame this paper as a direct follow-up rather than a fresh inquiry.
The torque result is the genuine contribution — domain-adapted Chronos-2 jumps from 0.42 to 0.69 F1 over CNN. The mechanism deserves more analysis beyond the brief alignment hypothesis offered.
The scale-dependency issue for hookload and its interaction with instance normalization is an important insight that deserves a dedicated ablation rather than a single paragraph.
Dataset provenance is unclear — it is not stated whether the labeled datasets here are the same as those in prior work, which directly affects how results should be interpreted.

Justification of Score
A solid practical paper with useful findings on torque anomaly detection and deployment tradeoffs. However, hookload results largely confirm prior work, Toto is treated asymmetrically, and the paper undersells its relationship to two closely related predecessors. Better positioning and stronger differentiation are needed.
Rating: 5 — Marginally below acceptance threshold
Confidence: 4 — Confident but not absolutely certain

---

### Official Review · Reviewer_NdmZ · 2026-05-21

**Rating:** 5
**Confidence:** 4

**Review:**

## Summary
The paper applies forecasting based TSFMs on an anomaly detection task for oil and gas drilling domain. Specifically, they finetune Toto and Chronos 2, and also train some other architectures from scratch. They show that Chronos 2 with a 2-stage domain specific finetuning (forecasting + classification) has strong results.

## Strengths
- Applies TSFMs for real world structured data domain (oil and gas drilling)

## Areas for Improvement
- Add stronger anomaly detection baselines
- Experiment with more TSFMs
- Refrain from making unqualified statements such as "The poor performance of Toto may indicate that encoder based TSFM models are better suited for dense segmentation than decoder-based forecasting models". This statement is purely speculative and not supported by the data/experiments.
- It is odd that only domain specific forecasting finetuning and random initialization experiments were only applied to Chronos 2 and not Toto.

## Justification of Score
The paper is mostly an application of TSFMs on a particular domain/dataset. While this direction is in scope of the workshop, it lacks reasonable experiments and baselines to draw conclusions about the models and approaches it takes. It would be interesting to learn whether a second stage domain specific pre-training phase does indeed improve downstream task performance, however, the way the experimented have been presented, unfortunately does not pass the bar.